



# First Light Multi-Frequency Observations with a G-band radar

Katia Lamer[1], Mariko Oue[2], Alessandro Battaglia[3,4,5], Richard J. Roy[6], Ken B. Cooper[6], Ranvir Dhillon[5] and Pavlos Kollias[1,2]

[1]Department of Environmental and Climate Sciences, Brookhaven National Laboratory, Upton, NY, USA
[2]Division of Atmospheric Sciences, Stony Brook University, NY, USA
[3]National Centre for Earth Observation, Leicester, UK
[4]Politecnico of Turin, Turin, Italy
[5]University of Leicester, Leicester, UK
[6]Jet Propulsion Laboratory, California Institute of Technology Pasadena, CA, USA

*Correspondence to*: Katia Lamer, (klamer@bnl.gov)

**Abstract** Observations collected during the 25-February-2020 deployment of the Vapor In-Cloud Profiling Radar at the Stony Brook Radar Observatory clearly demonstrate the potential of G-band radars for cloud and precipitation research, something that until now was only discussed in theory. The field experiment, which coordinated an X-, Ka, W- and G-band radar, revealed that the Ka-G pairing can generate differential reflectivity signal several decibels larger than the traditional Ka-W pairing underpinning an increased sensitivity to smaller amounts of liquid and ice water mass and sizes. The observations also showed that G-band signals experience non-Rayleigh scattering in regions where Ka- and W-band signal don't, thus demonstrating the potential of G-band radars for sizing sub-millimeter ice crystals and droplets. Observed peculiar radar reflectivity patterns also suggest that G-band radars could be used to gain insight into the melting behavior of small ice crystals.

G-band signal interpretation is challenging because attenuation and non-Rayleigh effects are typically intertwined. An ideal liquid-free period allowed us to use triple frequency Ka-W-G observations to test existing ice scattering libraries and the results raise questions on their comprehensiveness.

Overall, this work reinforces the importance of deploying radars with 1) sensitivity sufficient to detect small Rayleigh scatters at cloud top in order to derive estimates of path integrated hydrometeor attenuation, a key constraint for microphysical retrievals, 2) sensitivity sufficient to overcome liquid attenuation, to reveal the larger differential signals generated from using G-band as part of a multifrequency deployment, and 3) capable of monitoring atmospheric gases to reduce related uncertainty.

## 1 Introduction

Over the past 20 years, millimeter-wavelength radars have become the instrument of choice for the study of cloud and precipitation. Today, radars operating at 35- and 94-GHz frequencies are routinely operated at ground-based observatories (e.g., U.S. Department of Energy Atmospheric Radiation Measurement (ARM) user facilities (Stokes and Schwartz, 1994) and the Aerosol, Clouds and Trace gases Research Institute (ACTRIS; Pappalardo, 2018)) and





from a variety of ship-based and air-borne platforms (Kollias et al., 2007b). In space, the CloudSat 94-GHz cloud profiling radar has been operating since May 2006 (Stephens et al., 2002) and the Earth Cloud Aerosols and Radiation Explorer (EarthCARE), the first spaceborne Dopplerized cloud profiling radar, is expected to be launched in 2023 (Illingworth et al., 2015). Reasons for the popular use of millimeter-wavelength radars include that this frequency range is much more sensitive (in contrast to cm-wavelength radars) to cloud droplets and small ice crystals and that it allows for the collection of observations at excellent spatial resolution (~30m; Kollias et al., 2020a). Although non-Rayleigh scattering signatures in the radar Doppler spectra can be exploited for sizing large raindrops and snow (i.e., Mie notches techniques, Kollias et al., 2002), it remains challenging to extract quantitative information about the sizes and mass of small hydrometeors using observations from stand-alone single-frequency millimeter-wave radars. For the most part, challenges arise since signal at any one frequency experiences both attenuation (related to particle mass) and scattering (related to particle habit and size) making it nearly impossible to disentangle these effects.

Fortunately, attenuation and scattering of radar signals are frequency dependent such that they can be exploited to retrieve independent information about particle mass, habit, or size, depending on the character of scattering. For instance, this observations from two (or more) radar frequencies within the same scattering regime, but different absorption regime, can be combined to isolate differential attenuation signals useful for the retrieval of liquid water content (Hogan et al., 2005; Zhu et al., 2019). Alternatively, observations collected at two (or more) radar frequencies experiencing similar signal absorption, but differential scattering, can be combined to reveal information about ice crystal habit and size (Kneifel et al., 2015). That being said, modern multi-frequency pairings are limited because i) they rely on frequencies that experience little differential attenuation in liquid clouds causing larger liquid water content retrieval uncertainty and ii) they do not produce differential scattering signals for hydrometeors smaller than 800 $\mu m$, thus leaving a noticeable gap in our understanding of the microphysical properties of drizzle and small ice particles.

In response to these limitations, the research community has expressed an interest in developing radars operating at higher frequencies in the so-called G-band (110 – 300 GHz, Battaglia et al., 2014). Compared to a Ka-W (35-GHz-94-GHz) frequency pair, a Ka-G frequency pair should experience measurable differential attenuation at smaller water mass amounts and non-Rayleigh scattering at smaller particle sizes (e.g., Battaglia et al., 2014; Hogan and Illingworth, 1999; Lhermitte, 1988). What is more, a Ka-G frequency pair is expected to always produce differential signals larger than that of traditional pairs, thus increasing the resilience to noise and precision of hydrometeor mass or size retrievals. Other applications of G-band and submillimeter-wave radars come from the presence of a water vapor absorption line at 183 GHz. By tuning the radar frequency between positions of higher and lower absorption near a water vapor line, (e.g., 183 or 325 Ghz), G-band radars can be used to profile water vapor using the Differential Absorption Radar (DAR) technique (Battaglia and Kollias, 2019; Lebsock et al., 2015; Roy et al., 2018; Cooper et al., 2018).





Surprisingly, the first G-band radar built for meteorological applications was only developed in the late 1980's; McIntosh et al. (1988) designed a 215-GHz non-Dopplerized high-power Extended Interaction Klystron transmitter radar system and demonstrated that it was capable of making backscatter measurements from terrain targets at ranges of several kilometers under normal atmospheric conditions. Mead et al. (1989) attempted to use the system to characterize clouds and fog and realized that it did not possess sufficient sensitivity to detect clouds and light precipitation. It took 30 years before we saw the development of the next generation of G-band radars. In 2018, thanks to significant technological advancements in radar front ends, mixers and low-power wide-bandwidth solid state G-band sources, the Jet Propulsion Laboratory (JPL) developed a highly sensitive non-Dopplerized frequency-modulated continuous-wave (FMCW) G-band radar tunable from 167 to 174.8 GHz (i.e., DAR; Cooper et al., 2020; Roy et al., 2018). The system, named Vapor In-Cloud Profiling Radar (VIPR), was deployed during 7-days over April 2019 at the ARM Southern Great Plains facility to evaluate VIPR's ability to exploit differential absorption signatures to retrieved in-cloud humidity profiles (Roy et al., 2020). VIPR's retrievals were evaluated against coincident measurements from ARM water vapor sensors, with the primary comparison coming from frequently launched radiosondes. Furthermore, VIPR's integrated water vapor measurement capabilities in clear air columns were investigated by comparing with both radiosonde and Raman lidar profiles. These comparisons highlighted VIPR's ability to profile in-cloud water vapor with high resolution (< 200 m) and accuracy (RMSE < 1 g m$^{-3}$), especially within the planetary boundary layer. This deployment also helped identify regimes where VIPR's specific DAR channel locations (i.e., 167 and 174.8 GHz) resulted in retrieval biases stemming from frequency-dependent hydrometeor scattering properties. Shortly thereafter, VIPR was deployed aboard a DHC-6-300 aircraft from Twin Otter International Ltd for its first airborne measurement campaign in November 2019 and January 2020 (Roy et al, in prep).

VIPR was deployed again in February 2020 at the Stony Brook Radar Observatory (SBRO) to demonstrate the capability of G-band radars for characterizing rain, ice crystals, and snow. There, it collected observations alongside three radars operating respectively at 9.4, 35 and 94 GHz, thus providing first light multi-frequency radar observations including G-band. Here, we present the results of the quadruple-frequency radar field experiment that sampled a frontal system accompanied by pre-frontal cirrus clouds followed by ice transitioning into light warm rain. The presented work demonstrates the value of using a G-band radar as part of a multi-frequency radar observatory and underlines some important lessons learned and requirements needed for taking full advantage of G-band radar observations for cloud and precipitation microphysical studies.

## 2 Sensors and operations

The SBRO is a fenced-in facility located on the edge of Stony Brook University's commuter parking lot (40°53'50" N, 73°07'38" W). The SBRO is equipped with a W-band profiling radar, a Ka-band scanning polarimetric radar and, through a partnership with Raytheon, it also hosts two X-band dual-polarization low-power phased array radars (Kollias et al., 2018). In addition to these radar systems the SBRO is also equipped with a backscatter lidar, a long-





range scanning Doppler lidar as well as a surface flux system, and three Parsivel2 disdrometers. The observatory's equipment suite is completed by a sounding system, and a small drone with integrated meteorological sensors. When combined these systems have the ability to probe the atmosphere from surface to the top of the troposphere over horizontal scales of 20-40 km.


This section provides details specific to the operation of these systems during the deployment of the G-band VIPR radar on February 25th, 2020, beginning with a picture of the instrument layout during the field deployment (Fig. 1). Like the picture illustrates, all systems were installed in very close proximity in order to facilitate multi-frequency retrievals.


### 2.1 Vapor In-cloud Profiling Radar (VIPR)

VIPR is a first-of-its-kind solid-state G-band differential absorption radar (DAR). It's technical specifications are described in detail in Cooper et al. (2020) and Roy et al., (2020).


When it was deployed at the SBRO, VIPR transmitted 300 mW of power at 167 and 174.8 GHz, and was operated in frequency-modulated, continuous-wave (FMCW) mode with a chirp bandwidth of 10 MHz and corresponding range resolution of 15 m. With a single-pulse coherent integration time of 1 ms, VIPR realizes a noise-equivalent reflectivity of -40 dBZ at 1 km range. To reduce random noise from radar speckle, 2000 individual pulses are

incoherently averaged to form a single reflectivity profile, resulting in a temporal resolution of about 5 seconds. All observations reported here utilize the noise floor subtraction technique detailed in Roy et al. (2018), and any observations with signal-to-noise ratio below 0 dB have been removed from this analysis. For the multi-frequency analysis in this work, we only focus on the measurements at 167 GHz as this channel provides more cloud sampling than VIPR's higher frequency channel due to increased atmospheric absorption.


The radar was initially installed near, but outside a large shipping container such that it was able to collect atmospheric observations in vertically pointing mode. Following the onset of rain, VIPR's transmitter had to be turned off on a number of occasions to wipe water droplets off of the radar antenna (gaps seen in Fig. 8c). In some instances, we noted that strong radar returns from close-range rain caused an increase in the system noise floor of up to 20 dB

stemming from broadband phase noise in the transmitted signal (Cooper et al., 2020). At 20:41 UTC on 25 February, following the onset of heavier surface rain, VIPR was moved inside the adjacent container and pointed 40° off zenith. Note that these off-zenith observations were not analyzed as part of the current study.

### 2.2 W-band Profiling Radar (ROGER)


ROGER, named after late radar pioneer Roger Lhermitte, is a refurbished version of the W-band (94.8 GHz) radar previously integrated on the Center for Interdisciplinary Remotely Piloted Aircraft Studies Twin Otter aircraft (Mead





et al., 2003). ROGER is a single polarization 0.3° beam width Coherent Frequency Modulated Continuous Wave (CFMCW) radar with Doppler capability. Its range resolution is configurable between 5-150 m and it can detect targets up to a maximum range of up to 18.8 km. ROGER was refurbished by SBRO staff for ground-based vertically pointing operations in 2017. The effort involved building a new metal frame to hold the radar's two 24-inch parabolic dish antennas and all the CFMCW electronics as well as installing a server computer and power supplies.

During VIPR's deployment, ROGER was set to operate with a range gate spacing of 30-m and collected a full radar Doppler spectra every 4 sec achieving a sensitivity of roughly -30 dBZ at 1 km.

### 2.3 Ka-band Scanning Polarimetric Radar (KASPR)

KASPR is a mechanically scanning 0.3° beamwidth Ka-band fully polarimetric radar. Further details about KASPR can be found Kollias et al. (2020b).


For most of the VIPR deployment, until 21 UTC to be exact, KASPR was operating vertically pointing with 15 m range resolution and 13.6 km maximum range. It only transmitted H polarized wave and collected a full co-polar and cross-polar radar Doppler spectra every 1 sec achieving a sensitivity of roughly -42 dBZ at 1 km. Towards the end of the deployment, between 19:06-24:00 UTC, KASPR's vertically pointing observations were supplemented every 5-

minutes by a 15° elevation plan point indicator scan (PPI) and a hemispheric range-height indicator scan (HS-RHI; Kollias et al., 2014). Both scan types were designed to collected dual polarization observations at 45 m range resolution for a 30 km range. Note that the scanning observations were not analyzed as part of the current study.

### 2.4 X-band dual-pol phased array radar (SKYLER)


SKYLER is a dual-polarization X-band low-power phased-array radar with an antenna beamwidth of 1.98° in azimuth and 2.1° in elevation at boresight. SKYLER's full range of capabilities are described in Kollias et al. (2020b).

During VIPR's deployment, SKYLER was only operated between 18:00-24:00 UTC. SKYLER was mounted on a

rotation table installed on a mobile truck's flatbed oriented facing upward to enable the collection of vertically pointing observations. SKYLER was set to operate with a 2 μs pulse, 48 m range gate spacing with maximum range of 9.85 km. For collection of observations at 1-s time resolution, SKYLER was able to achieve a sensitivity of roughly +15 dBZ at 1km.

Because SKYLER's receiver blanking parameters were incorrectly set, its reflectivity observations collected below 1.25 km are biased low (hashed region on Fig. 8a). Knowing that this bias could be corrected for, we elected to display these observations, but only performed quantitative retrievals using SKYLER observations collected above 1.25 km.





### 2.5 Ancillary Measurements

One of the SBRO Parsivel2 laser optical disdrometers was operating during the VIPR's deployment. Vendor provided algorithms were used to classify the Parsivel2 drop observations into 32 separate size and velocity classes every 1 minute. In this work, Parsivel2 observations are mainly used for conducting power calibration of all four radars.

The National Weather Service (NWS) performs balloon-born radiosonde measurements twice a day (00:00 UTC and 12:00 UTC) from the Brookhaven National Laboratory campus in Upton NY, 22 km east of the SBRO location. On February 25, 2020, SBRO staff and Stony Brook University students also launched two GRAW DFM-90 radiosondes at 01:46 UTC and 15:44 UTC directly from the SBRO.

A Stream Line XR Doppler LiDAR and a Lufft CMH 15k backscatter lidar were also operated during the field experiment. The Doppler lidar was set to operate at 60 m range resolution and 1 sec temporal resolution, providing estimates of air motion in the subcloud layer (not analyzed as part of the current study) while the backscatter lidar was set to operate with a 15 m range resolution and 15 s temporal resolution for monitoring the location of liquid layers.

## 3 Radar data post-processing

Before they can be used to gain insight on atmospheric liquid and/or ice, high-frequency radar measurements must be post-processed to remove signal attenuation caused by atmospheric gases. Also, and especially in the content of multi-frequency analysis, radar signals should be calibrated to improve the accuracy of quantitative retrievals. This section describes the steps used to post-process and calibrate the radar observations collected by the VIPR, ROGER, KASPR and SKYLER radar and how these corrected observations are combined to conduct a multi-frequency analysis.

### 3.1 Gaseous attenuation correction

When thermodynamic information is available, radio wave propagation models can be used to estimate radar signal attenuation by atmospheric gases. Here we use the MPM93 model, an updated version of the millimeter-wave propagation model described by (Liebe, 1985; Liebe et al., 1993), to compute two-way gas attenuation of X-, Ka-, W- and G-band signals for the conditions that occurred at 12:00 UTC and 15:44 UTC on February 25[th], 2020 when two radiosondes were launched. Figures 2a and 2b show the profiles of temperature, dew point temperature and humidity recorded at the NWS site 22 km east of SBRO at 12:00 UTC and at the SBRO at 15:44 UTC. The two-way gas attenuation profiles depicted in Fig. 2c confirm that millimeter radar signals, particularly at G-band, experience non-negligeable gas attenuation. For this particular mid-latitudinal winter case, we estimate two-way gas attenuation at 11 km to reach ~0.1 dB at X-band, ~0.5 dB at Ka-band, ~2.0 dB at W-band and 10.0 dB at G-band. The large variability in gas attenuation from frequency to frequency, especially near water vapor absorption lines, is what allows DAR techniques to be used for water vapor profiling. On the upside the notable magnitude of the gas attenuation at higher-



frequencies (i.e., W-band but even more so G-band) makes them ideal frequencies to use for such application. On the downside, significant gas attenuation hinders the sensitivity of high frequency radars to clouds and light precipitation.

Since the following analysis focuses on quantifying hydrometer properties, we correct all radar signals for two-way gas attenuation using the profiles derived above. The profiles estimated using the 12:00 UTC sounding are used to correct radar measurements collected before 13:52 UTC, while the ones estimated using the 15:44 UTC sounding are used to correct the rest of the radar measurements. The variability between the consecutive profiles can be used to get a sense of the uncertainty associated with using only two soundings to correct the daylong radar dataset.


### 3.2. Radar reflectivity calibration

On February 24, 2020 (one day before the official field experiment), VIPR's calibration was verified using the methodology described in Roy et al. (2020); the exercise required hanging a small calibration sphere between two
light posts roughly 200 m from the SBRO. KASPR's calibration is similarly checked twice a year by SBRO staff using a corner reflector located 300 m away from the SBRO.

SKYLER, ROGER and KASPR measurements are also sporadically calibrated using Parsivel2 measurement collected during rain episodes following a standard calibration technique similar to that described in Chandrasekar et al. (2015)
and Kollias et al. (2019). In short, the Parsivel2 disdrometer particle size distribution (PSD) measurements are used as input to a T-matrix scattering algorithm (Mishchenko et al., 1996) that estimates the hydrometeors radar reflectivity for radar frequencies of interest. The idea is then to compare the disdrometer-derived radar reflectivity estimates to the reflectivity observed by the radar at the same height and use their difference to calibrate the radar measurement across the entire atmospheric column. Additional steps arise from the fact that radars generally do not collect
measurements down to surface level where disdrometers are located. The several hundred-meter path between these measurements results in three sources of systematic calibration error: 1) radar signal attenuation by atmospheric gases present in the path, 2) radar signal attenuation by the raindrops present in the path and 3) a time lag reflecting the time it takes raindrops to fall from the observed height to the surface. All these effects must be corrected for before comparing radar reflectivity observed at the lowest observation height to the disdrometer-derived radar reflectivity
estimates. The technique described in the previous section can be used to correct for gas attenuation along the path. Liquid attenuation can be estimated using a T-matrix scattering algorithm and Parsivel2 PSD measurements assuming that the PSD remains constant along the path between the radar's lowest observation height and the surface (355 m for KASPR, ROGER and VIPR is and 1,250 m for SKYLER). Then, time-series analysis can be used to correct for the time lag between the corrected radar reflectivity (from the radar at the lowest observation height) and the
disdrometer-derived radar reflectivity estimates.

For this analysis, Parsivel2 measurements collected between 17:53-24:00 UTC are used to calibrate the measurements from all four radars. Because of uncertainties in Parsivel2 PSD measurements (Tokay et al., 2014), only rain PSDs





with mean diameter greater or equal to 0.6 mm are considered for the calibration procedure (refer to Fig. 3a for details).
The median difference between the disdrometer-derived and radar-derived (corrected for gas and liquid attenuation and time-lag) radar reflectivity over the rain episodes was used to calibrate the entire radar data record collected on that day. The resulting calibration coefficients amount to -8.1, 0.2, -2.3, and 1.3 dB for SKYLER, KASPR, ROGER, and VIPR respectively. The small calibration coefficients found for VIPR and KASPR also suggest that the target and corner reflector calibration procedures performed for these radars were reasonably effective.


### 3.3 Multi-frequency analysis

Ideally, multi-frequency analysis would be performed using perfectly time-matched and volume matched observations in order to be able to attribute any signal differential to the properties of the hydrometeor population. Unfortunately,
previous work has shown that perfectly matching radar observations is extremely challenging even for radars installed on the same pedestal (Kollias et al., 2014). Observation volume differences unavoidably occur as a result of using different radar frequencies, which require the use of different transmitting configurations such as pulse width, pulse repetition frequency, and number of samples per integration. Temporal and vertical averaging of radar data on a common grid has been used in an attempt to alleviate radar observation mismatching.


Here we co-gridded the post-processed radar observations from all four radars on a joint 15 m, 4 sec resolution grid. The gridded observations are subsequently averaged in time in 60-s increments to reduce noise. The denoised radar observations are used to estimate the dual wavelength ratio ($\text{DWR}_{A-B} = dBZ_A - dBZ_B$, in dB) for three pairs of observed radar reflectivity (Ka-W, Ka-G and W-G).


### 4 Key findings from the multi-frequency radar deployment

On 25 February 2020, following the movement of a surface trough and associated low-pressure system, a stationary front established itself over the SBRO. The four profiling radar systems and the two lidar systems operating at the
time observed the transition from pre-frontal cirrus to rain associated with this system. The following sections discuss key findings attributable to the deployment of a G-band radar as part of a multi-frequency radar deployment in these two weather regimes.

### 4.1 Using G-band for ice crystals sizing and habit characterization


The radar and lidar observations displayed in Fig. 4 reveal that a deck of prefrontal cirrus clouds, whose top extended near 9-10 km, advected over the observatory between 7:00 and 10:00 UTC. Observations from KASPR, ROGER and VIPR show that the thickness of the cirrus layer varied over time between ~2 and 6 km in depth. In the lowest part of the cloud layer, moderate lidar backscatter signals (~$10^{-4.2}$ m$^{-1}$ sr$^{-1}$) suggest the presence of high number concentrations
of small particles. Thin bands of high lidar backscatter signals (~$10^{-3}$ m$^{-1}$ sr$^{-1}$) near 3.0 and 4.0 km support the idea



that supercooled liquid layers were also present in the lowest part of this cloud system certainly in the earlier and later parts of the period, and likely over the entire period (Fig. 4e). If so, interaction with supercooled liquid could have influenced the ice particle growth processes in the atmospheric column. The mean Doppler velocity recorded by KASPR offers additional insights into the complex dynamical and microphysical structure of the observed layer (Fig.

4d). The signature of a gravity wave with an air velocity of 0.3-0.4 m s$^{-1}$ and a period of 5-6 min is clearly evident throughout the hydrometeor layer. Several, higher frequency dynamical features are also identifiable in what appears like mammatus clouds features in the lowest 2 km of the system between 8:45-9:15 UTC.

Differences in radar reflectivity measured by the Ka- (Fig. 4a), W- (Fig. 4b) and G- (Fig. 4a) band radars are a direct result of differences in signal attenuation and scattering, which can be best visualized in dual-wavelength ratio (DWR) space. Figure 5 shows DWR estimated using the traditional Ka-W pair (panel a), the Ka-G pair (panel b) and the W-G pair (panel c). These first light DWR observations involving G-band confirm all the advantages predicted by scattering theory.

Focusing below ~4.5 km, we observe that in contrast to the Ka-W pair (Fig. 5a), the frequency pairs with G-band (Fig. 5b-c) indeed experiences larger differential signals for the same hydrometeor population; for the case observed, the DWR profile shown in Fig. 6b allows us to estimate that this gain was as large as ~4 dB for the Ka-G pair in comparison to the Ka-W pair. This increased dynamic range in DWRs corresponds to an increased sensitivity in the transfer function between DWRs and microphysical properties. This underpins the value of using frequency pairs farther apart

in the frequency spectrum not only to mitigate the impact of possible noise when retrieving the size of smaller particles or lower water mass amounts but also to increase retrieval precision. Finally, observations collected above 4.5 km reveal the G-band's strength in small particle regimes. In this region, absence of Ka-W differential signal (i.e., DWR = 0 dB) suggests the presence of Rayleigh targets (Tridon et al., 2020), which for these frequencies correspond to ice populations with PSDs of mass-weighted mean diameter smaller than ~1 mm (Tridon et al., 2019). While Ka- and W-

band signals lack sufficient differential scattering to gain further information about such small ice crystals, our observations suggest that G-band signal can; DWR estimated using G-band show differential signals on the order of a few decibels across most the layer (see Fig 5b-c and Fig. 6b).

Interpreting and performing retrievals from DWR observations always requires considering the interplay of signal attenuation and non-Rayleigh scattering (Tridon et al., 2013). Observations collected during the period around 8:42 UTC highlight this important limitation of DWR analysis targeting the characterization of ice crystals. The lack of converge at 0 dB in the profile extracted at 8:42 UTC suggests the presence of considerable water condensate mass in the atmospheric column (Fig. 6d). Backscatter lidar observations do allude to the presence of liquid layers (of unknown depth) over that period (Fig. 4e). Tridon et al. (2020) suggest that if DWR reaches a constant value with height (a.k.a.

a Rayleigh plateau), the DWR of this plateau can be used to estimate integrated water condensate mass within the layer. In this particular profile, the Ka-W-band pair reached a clear Rayleigh plateau at 5 km showing a 1 dB DWR loss to hydrometeor attenuation. We argue that both the Ka-G pair and the W-G pair also reached a Rayleigh plateau



near 6.8 km showing in the neighborhood of 3.5 dB DWR loss to hydrometeor attenuation. This signal could be qualified as being the first quantitative hydrometeor mass signal recorded at G-band. Because ice and snow attenuation considerably increase when moving from the W- to the G-band reaching one-way values of 0.9, 2.5, and 8.7 dB $m^2 kg^{-1}$ at 96, 140, and 225 GHz (Tridon et al., 2020; Nemarich et al., 1988), the DWR plateau for the G-band pairs is affected by both the liquid water path and the ice water path. On the other hand, the DWR plateau for the Ka-W pair is mainly driven by the liquid water path. We believe that the shallowness of the W-G band plateau results from the limited sensitivity of ROGER, which is likely insufficient to detect additional Rayleigh targets populating the top of the ice cloud. This observation supports the need for developing highly sensitive radars when targeting small (in size) hydrometeor populations. Unfortunately, because millimeter-wavelength radar signals alone cannot be used to precisely distribute the retrieved water path across the atmospheric column, non-Dopplerized DWR observations in mixed-phased clouds cannot be disentangled to isolate the non-Rayleigh signals required for sizing and identifying ice crystal habit leaving yet again a gap in our understanding.

The DWR profile shown in Fig. 6b taken from observations collected at 8:00 UTC shows a contrasting situation where G-band signals can be directly used for ice microphysical retrievals. In that profile DWR is seen to converge to 0 dB such that differential signal across the column can be interpreted from resulting exclusively from non-Rayleigh scattering. Under such conditions DWR can be related to ice crystal size given the proper ice scattering library. Kneifel et al. (2015) initially proposed using $DWR_{X-Ka}$ versus $DWR_{Ka-W}$ diagrams to identify ice particle types from multi-wavelength radar observations. Recently, it has become evident that details of the PSDs and unaccounted attenuation complicate the analysis of such diagrams that must be interpreted with caution (Battaglia et al., 2020). Figure 7 shows $DWR_{Ka-W}$ versus $DWR_{W-G}$ diagrams for two periods encompassing the profiles described above. Overlaid are $DWR_{Ka-W}$-$DWR_{W-G}$ estimated using self-similar-Rayleigh-Gans approximation and different particle type models and PSD. Unrimed aggregates are represented using the mass-diameter relationships from Hogan and Westbrook (2014) (hereafter HW) and aggregates with increasing riming for different amounts of liquid water path from 0 to 2 $kgm^{-2}$ are represented using two particle models with unique mass-diameter relationships from Leinonen and Szyrmer (2015) (hereafter LS15A and LS15B). PSDs are represented using a gamma function with a shape parameter (µ) of either 0 or 4. The idea is to use overlap between the observed and estimated DWR-DWR to gain information about particle habit.

The first period (7:45 – 8:12 UTC) depicted in Fig. 7a corresponds to the period we established had little to no liquid water and presented a high DWR slanted feature (referring back to Fig. 5). Absence of water attenuation can also be confirmed in DWR-DWR space by a clustering of observations collected between 5.75-7.00 km near the 0,0 point (depicted by the contours on Fig. 7a). For this particular period, a 0.5 dB offset is seen suggesting that a slight adjustment should be made to the observed DWR before they can be used to infer particle habit. However, such a small adjustment would likely be insufficient to produce DWR-DWR signals matching the scattering models used. This suggests that the particles observed are not represented the scattering libraries used and calls for further research. The scattering models that are closest to the observed values are those for unrimed particles (yellow and magenta



lines). Attempting to further characterize these ice crystals, we note that the sounding reported a temperature in the region of roughly -15°C and relatively humidity of roughly 80 % (referring to Fig. 2). Under such thermodynamic conditions, high $DWR_{Ka-w}$ are typically associated with the presence of dendritic crystals and aggregates (e.g., (Bechini et al., 2013;Andrić et al., 2013)). From changes in height of the $DWR_{Ka-w}$ peaks over the period, we estimate the fall speed of the ice particles to be roughly 1.9 m s$^{-1}$. The large DWR values and the low terminal velocity suggest

the presence of large and fluffy, unrimed particles (Locatelli and Hobbs, 1974).

The second period (8:12-9:12 UTC) depicted in Fig. 7b corresponds to the period containing non-negligible attenuation by water condensates. This period also presents a broad high DWR area between 2 and 5.5 km altitude (referring back to Fig. 5). The offset from 0,0 $DWR_{Ka-w}$-$DWR_{W-G}$ of observations collected between 5.75-7.00km can

also be used to confirm the presence of water condensates (depicted by the contours on Fig. 7b). Although tempting, it is not possible to directly interpret this DWR-DWR diagram since details about the vertical distribution of the liquid and ice water content is not known and as such attenuation cannot be accurately corrected for. From changes in height of the $DWR_{Ka-w}$ peak over the period, we estimate the fall speed of the ice particles to be roughly 3.7 m s$^{-1}$.

**4.2 Using G-band for characterizing melting and sizing submillimeter drizzle droplets**

The radar observations displayed in Fig. 8 show the light surface rain episode that occurred following the frontal passage between 18:00 and 18:30 UTC. Observations from KASPR allow us to establish that the cloud sustaining the rain extended up to 8 km. The bright band observed by all radars, although notably different, is suggestive of a

transition from ice particle to liquid water near 2 km. This idea is substantiated by radiosonde reports that place the 0-degree isotherm near 2 km (Fig. 2a). Surface disdrometer measurements indicate that rainfall rate at the surface varied reaching up to 2.1 mm hr$^{-1}$ during the period (Fig. 2a within the limits of period 3). From time-lag estimates performed as part of the calibration procedure, we estimate that the rain drop fall speeds ranged from 3 to 6 m s$^{-1}$. These estimates are consistent with KASPR mean Doppler velocity measurement made during the period (not shown).


Difference in radar reflectivity measured by the X- (Fig 8a), Ka- (Fig. 8b), W- (Fig. 8c) and G- (Fig. 8d) band radars during the period and specifically at 18:07 UTC (Fig. 9a) are a direct result of difference in signal attenuation and scattering.

Differential signal scattering explains the progressive reduction in the overall radar reflectivity factor measured by the X-band SKYLER, Ka-band KASPR, W-band ROGER and G-band VIPR. For raindrop sizes typical of such light precipitating systems (see Fig. 2a for estimates from the disdrometer), it can be safely assumed that X-band waves with their 3.2 cm wavelength ($\lambda$) experience Rayleigh scattering. In the Rayleigh scattering regime, radar backscattering cross section ($\sigma_b$) is proportional to $D^6/\lambda^4$ where $D$ is particle diameter. Because wavelength is much

larger than particle diameter $\sigma_b$ tends to be very small in that regime. That being said, the radar reflectivity factor ($Z$), which was designed to compensate for the wavelength dependency, can acquire very high values in that scattering



regime ($Z \sim D^6$). In contrast to X-band signals, Ka-, W- and G-band signals are expected to experience both Rayleigh scattering (for drops smaller in size relative to the wavelength) and non-Rayleigh scattering (for drops larger in size relative to the wavelength. In the non-Rayleigh scattering regime, $\sigma_b$ does not monotonically increase with $D^6$ but rather follows a lower power of quasi-periodic form with exponential damping of the oscillation (Fig. 4 of Kollias et al., (2007a)). As a result, although in non-Rayleigh scattering $\sigma_b$ acquires much higher values than those in Rayleigh scattering, the reported radar reflectivity factor during non-Rayleigh conditions is lower. Variations in each of the radars' "dominant" drop population (i.e., the largest drop size behaving as a Rayleigh scatterer), also explains variations in the observed radar bright band. SKYLER, like a typical centimeter wavelength radar, observed a bright band marked by clear boundaries at both the top and the bottom. Previous work has associated the top boundary to the initial melting of ice crystals, the peak to the stage where the largest ice crystals have acquired a wet coating, and finally the bottom boundary to the collapsing of the largest ice crystal into raindrops and to particle flux divergence where the largest raindrops have first rapidly fallen out of the melting layer leaving the smaller less reflective ones behind (Fabry and Zawadzki, 1995). In contrast, because millimeter-wavelength signals are dominated more by raindrops than large melted ice crystals, KASPR and ROGER did not observe such a strong bright band peak and neither a clear bottom boundary (Kollias and Albrecht, 2005). Also, the top boundary of their bright band was generally slightly higher as result of observing the melting of small ice crystals which occurs before the melting of larger ones. Interestingly, observations collected by the VIPR reveal a well-defined bright band at G-band frequency. The top boundary of the VIPR's bright band is slightly higher compared to that of the W-band, suggesting it is driven by the melting of even smaller ice crystals. We also note that the bottom of VIPR's the bright band is higher than that of SKYLER. Part of this discrepancy could be explained by the fact that SKYLER has a much larger range resolution than VIPR (300 m vs. 15 m). Past studies have related the depth of the bright band in centimeter-wavelength radars to the depth of the layer below the 0°C isotherm height required for the complete melting of the largest snowflakes. Due to strong non-Rayleigh scattering conditions in the melting layer, the large snowflakes do not contribute to the observed VIPR's bright band signature. This is a plausible explanation for its shallow depth.

Although G-band signals should allow for sizing smaller raindrops because they experience non-Rayleigh scattering at smaller droplets sizes (compared to longer frequencies), one must remember that G-band signals also experience non-negligeable liquid attenuation. Theoretical calculations suggest that extinction coefficients at 94- and 220-GHz rapidly increase for particles with size up to $D_m \approx 1$ mm and 0.4 mm, respectively, and then steadily decrease as a function of $D_m$ (Lhermitte, 1990). For the duration of the observed rain event, we estimate (from disdrometer PSD measurements) that two-way liquid attenuation of the G-band signal varied from 0 to 10 dB (Fig3 c). While non-negligible, this value is only about 2.2 times (in linear scale) higher than that experienced by a W-band radar like ROGER or the CloudSat-CPR (Fig. 3c; Battaglia et al., 2014). As seen in Fig 8 both ROGER and VIPR were both equally able to penetrate through the 2 km thick rain layer and detect a large portion of the cloud aloft (Fig. 8c and d respectively). The fact that VIPR and ROGER could not observe the cloud top speaks to the importance of operating highly sensitive G-band and W-band radars especially if they are meant to document the properties of liquid





precipitating clouds. The other fact that SKYLER could also not observe the cloud top also speaks to the importance of operating sensitive X-band radars for cloud studies (liquid attenuation not being an issue at X-band).

Like we saw in ice clouds, large $DWR_{Ka-G}$ and $DWR_{W-G}$ were measured during the rain event (Fig 9b); in this example profile collected at 18:07 UTC $DWR_{Ka-G}$ reached values as high as 30 dB. Interpreting these signals requires separating the contributions of liquid attenuation and non-Rayleigh scattering. In regimes with large $D_m$ (> 1 mm), similar liquid attenuation at W- and G-band should allow for the interpretation of $DWR_{W-G}$ signals in terms of differential scattering

caused by liquid drops (that is when gas attenuation has been corrected for). Such interpretation is arguably more challenging using the Ka-W or the Ka-G frequency pair (Matrosov, 2005).

**5 Conclusions**

Several observational gaps in cloud and precipitation remote sensing observations still exist especially at the mid and high latitudes (Battaglia et al., 2020). Radars at frequency above 100 GHz are now technologically feasible as proved by the VIPR system, recently built by JPL. This work presents multi-frequency (X-, Ka-, W- and G-band) radar observations from a field experiment at the Stony Brook Radar Observatory (SBRO). Albeit short, the field experiment provided a long-sought-after first light demonstration of the potential of multiwavelength radar observations that

include G-band for the characterization of ice crystals, snow and rain. Besides confirming expectations derived from scattering theory, the field experiment revealed a number of considerations relevant to the deployment of G-band systems.

1)   The observations clearly demonstrate that G-band radars can be made sensitive enough to probe clouds and light

precipitation and that in spite of the strong water vapor attenuation occurring at this frequency. The large sensitivity of G-band radars can in part be explained by improvements in radar gain with increased frequency; all else equal, for a fixed aperture size, radar sensitivity improves by 24 dB going from 10 to 170 GHz.

2)   Since G-band signals are especially prone to attenuation by water vapor, we recommend that G-band radars

targeting the characterization of clouds and precipitation should have differential absorption capabilities in order to avoid confounding effects due to water vapor attenuation. This could be achieved through the use of interlaced pulses whose frequency would range around a water vapor absorption line. The exact frequency range should ideally be tuned to the specific water vapor condition like proposed in Roy et al. (2020), Cooper et al. (2020) and Battaglia and Kollias (2019).


3)   The observations presented here reinforce the idea that the sensitivity of all the radar systems involved in future multi-wavelength radar studies should be sufficient to allow the detection of the Rayleigh plateau near the top of ice clouds; that is necessary to ensure that we have a robust estimation of the differential (dual-wavelength) path integrated liquid attenuation (Tridon et al., 2020). For rain studies as well, G-band radar sensitivity should be





large enough to allow signals to penetrate through the rain shaft and that despite attenuation by liquid water reaching several dBs. Considering gaseous and hydrometeor attenuation, the radar systems should have a sensitivity of -20 dBZ at 10 km altitude (i.e., -40 dBZ at 1 km) after 1-s signal integration. In the present study, the radars deployed generally meet this sensitivity criteria.

4)    The observation collected during this experiment confirm that the Ka-G pair generates the strongest differential reflectivity signal, with observed values of DWR reaching up to 13 dB in ice regions; 4 dB larger than traditionally Ka-W pairs. The increased differential signal should allow for increase retrieval confidence, especially in low liquid water content regions and/or for small particle sizes.

5)    The steep $DWR_{Ka-G}$ gradients observed support the idea that Ka-G differential signals are more sensitive to incremental changes in particle size thus allowing for more precise quantitative retrievals compared to those achievable using a Ka-W pair.

    6)    In the absence of Ka-W differential signals, observations of non-Rayleigh scattering differential signals at Ka-G
and W-G demonstrates the potential of G-band radars for sizing smaller ice particles.

    7)    An ideal case observed during the field experiment allowed us to investigate ice crystal habit. DWR-DWR observed by the Ka-W-G trio were compared to estimates made using several scattering libraries. No matches were found suggesting that existing scattering libraries may still not provide a comprehensive picture of the
scattering properties of naturally occurring ice crystals. Radiosonde measurements and DWR signatures suggest that the ice crystal observed were likely large in size and unrimed. Additional triple frequency observations including G-band would help confirm this finding, which, if correct, should motivate the development of additional ice scattering libraries.

8)    Observations collected during a melting event suggest that G-band radars can detect radar bright bands. The character of this bright band is likely indicative of the melting behavior of smaller ice crystals.

    9)    In rain, the G-band radar reflectivity values are several orders of magnitude lower than those measured by the W-band, Ka-band and X-band radar systems creating measurable DWR signal. Interpreting these differential signals
may be challenging because they result from both differential scattering and attenuation. In large particle regimes where W- and G-band signals experience similar attenuation by liquid attenuation, $DWR_{W-G}$ should provide information more closely related to the mass-weighted diameter of the particle size distribution. Ideally full Doppler spectra capabilities should be added to G-band radars. Especially for applications in rain and mixed-phase clouds, Doppler capability would allow for application of spectral ratio techniques like proposed in Tridon
et al., (2013).



Longer datasets with similar measurement capabilities are needed to fully assess the potential and challenges associated with using non-Dopplerized G-band radar observations for the study of clouds and precipitation systems. Such observations can in turn be used to raise the technology and science readiness levels of space-borne G-band

systems. Because of their reversed observation geometry, G-band radar signals from an above-cloud vantage point should suffer from less signal attenuation than ground-based systems thus requiring a lower sensitivity to collect similar observations; that is because water vapor and rain are typically concentrated in the lowest part of the atmosphere, which spaceborne G-band radar signals will encounter last. The reduced signal attenuation should drive a less stringent sensitivity requirement (-20 dBZ in the troposphere after signal integration of ½ of the radar footprint).


*Data availability.* The datasets collected at the SBRO during the field experiment will be made publicly available. The NWS sounding data is available at https://www.spc.noaa.gov/exper/soundings/.

*Author contribution.* K. Lamer, M. Oue, A. Battaglia, R. Roy, K. Cooper and P. Kollias were actively involved in

the field experiment; K. Lamer operated the Doppler lidar, M. Oue operated the KASPR, and R. Roy and K. Cooper together operated VIPR. M. Oue and R. Dhillon performed initial data exploration work. P. Kollias finalized the data analysis. A. Battaglia and P. Kollias's input were instrumental in interpreting the radar signatures observed. K. Lamer lead the writing of the final version of this manuscript. All members of the team reviewed and added to this final version.


*Acknowledgements.* K. Lamer was supported by Brookhaven National Laboratory LDRD #20-002 EE/EBNN. M. Oue and P. Kollias were supported by the National Science Foundation award #1841246 entitled: "Collaborative proposal: Studies of the microphysical processes in ice and mixed-phase clouds and precipitation using multiparameter radar observations combined with cloud modeling". A. Battaglia and R. Dhillon were supported by

UK-CEOI under the Grace project. The research by R. Roy and K. Cooper was carried out at the Jet Propulsion Laboratory, California Institute of Technology, under a contract with the National Aeronautics and Space Administration (80NM0018D0004). This research also used the Advanced Leicester Information and Computational Environment (ALICE) High Performance Computing Facility at the University of Leicester, UK. We would like to thank the Brookhaven National Laboratory staff and Stony Brook University students who assisted during the field

experiment; special thanks go to Edward Luke for operating SKYLER and to Zeen Zhu, Samantha Nebylitsa, Jacob Segall, and Kristofer Tuftedal for launching radiosondes from the SBRO.

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





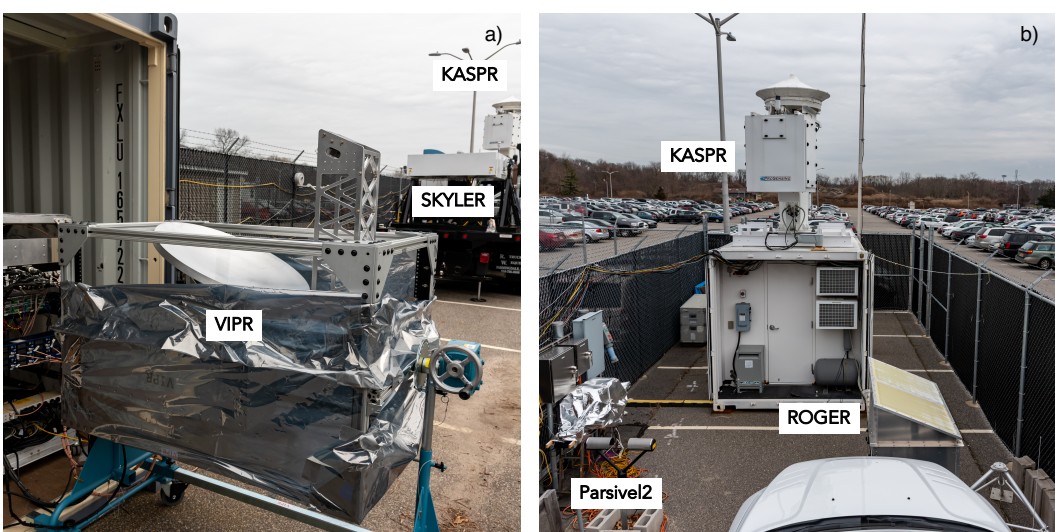

**Figure 1: a) Picture of the VIPR G-band radar system when it was deployed at the Stony Brook Radar Observatory. Also deployed at the observatory was the a truck-mounted X-band phased-array named SKYLER (visible in a), the container-mounted parabolic-dish Ka-band radar named KASPR (visible in a and b), the FMCW W-band radar named ROGER (visible in b) and a Parsivel2 disdrometer (visible b).**

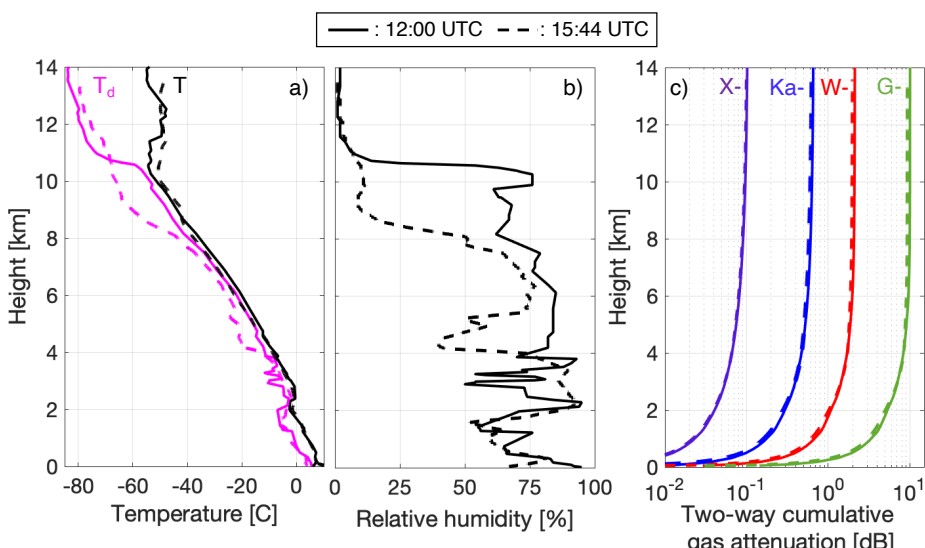

**Figure 2: From sounding observations collected on 25 February 2020 at 12:00 UTC at the NWS Upton site (22 km east of SBRO; solid lines) and at 15:44 UTC from the SBRO (dashed lines); Profile of a) temperature (black) and dew-point temperature (magenta), b) relative humidity, c) two-way water vapor attenuation at X-band (purple), Ka-band (blue), W-band (red), and G-band (green).**





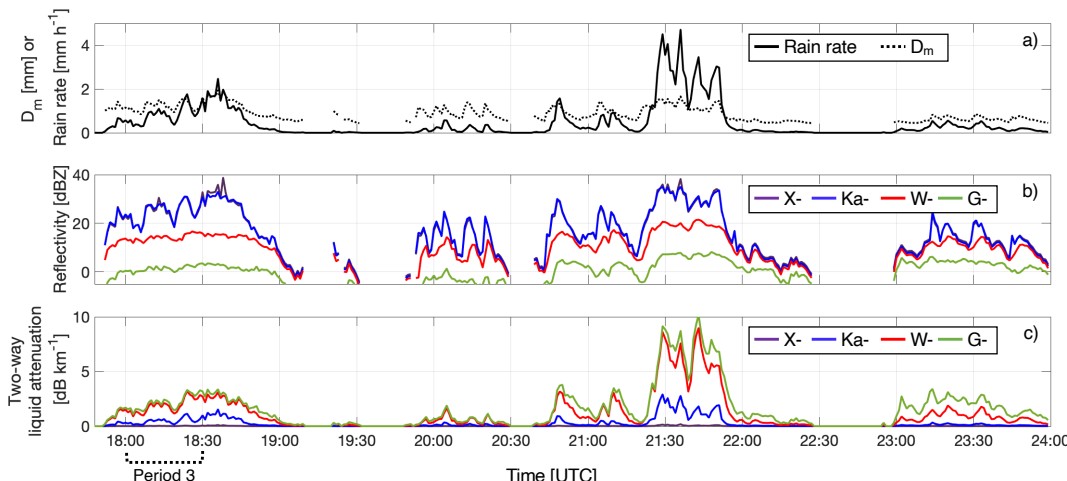

**Figure 3: Based on measurements from the Parsivel2 disdrometer collected on 25 February 2020, time series of estimated a) particle size distribution mass-weighted mean diameter ($D_m$; dotted line) and rain rate (solid line), b) radar reflectivity, and c) two-way liquid attenuation for X-band (purple), Ka-band (blue), W-band (red) and G-band (green).**




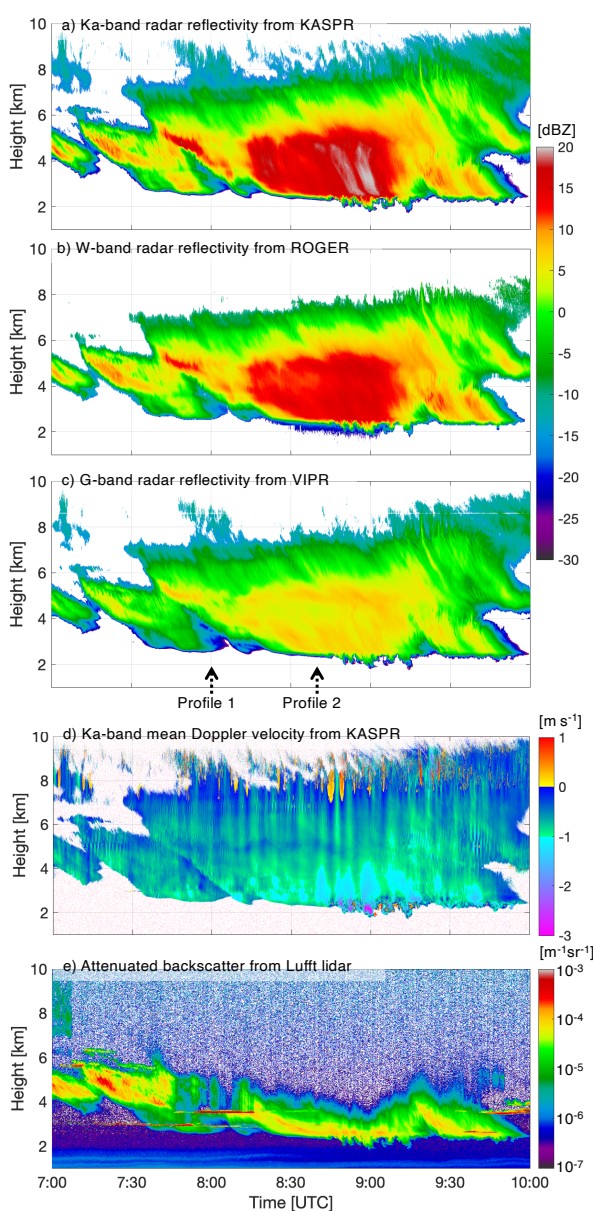

**Figure 4: Time-height of radar reflectivity measured by the a) KASPR, b) ROGER and c) VIPR, between 7:00 and 10:00 UTC on 25 February 2020. The arrows in (c) points to the time of the profiles displayed in Fig. 6. Also shown are a time-height of d) mean Doppler velocity measured by the KASPR (positive values indicate upward motion) and e) attenuated backscatter measured by the Lufft lidar.**





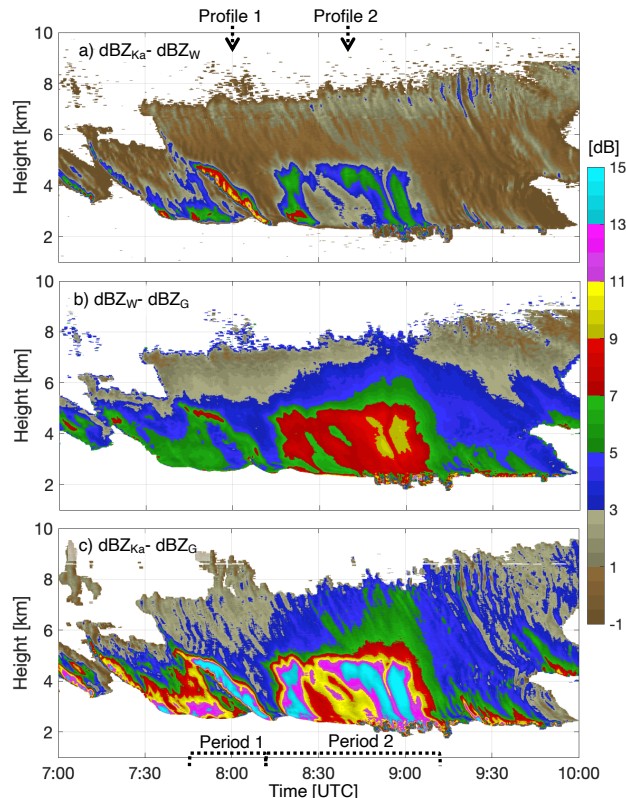

**Figure 5: Time-height of dual wavelength ratio from a) the Ka-W pair, b) the W-G pair and c) the Ka-G pair estimated between 7:00 and 10:00 UTC on 25 February 2020 (same date and time as in Fig. 4). The arrows in (a) points to the time of the profiles displayed in Fig. 6, while the periods outlined in (c) are the focus of Fig. 7.**






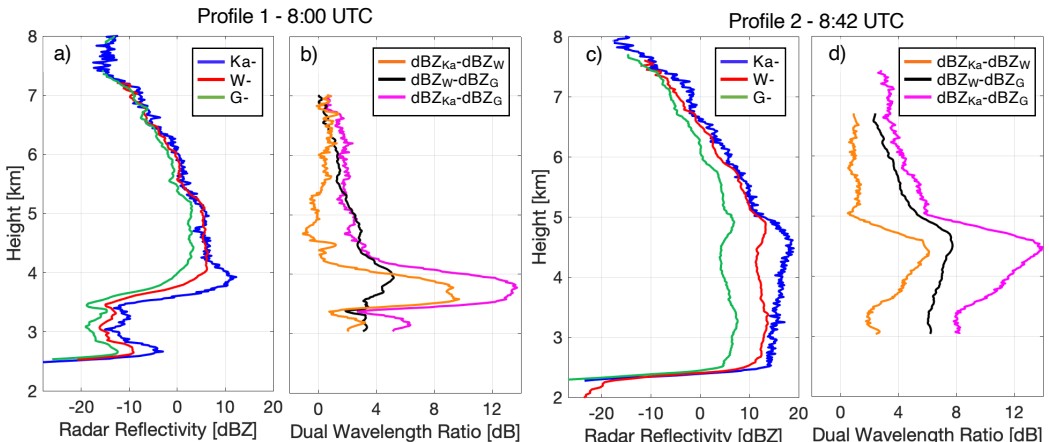

**Figure 6: Profiles taken at 8:00 UTC during the ice cloud period, a) radar reflectivity measured by VIPR (G-band, green), ROGER (W-band, red), and KASPR (Ka-band, blue) and b) associated dual-wavelength ratio from the Ka-W pair (orange), the W-G pair (black) and the Ka-G pair (magenta). c) and d) show the same information for the profile taken at 8:42 UTC.**

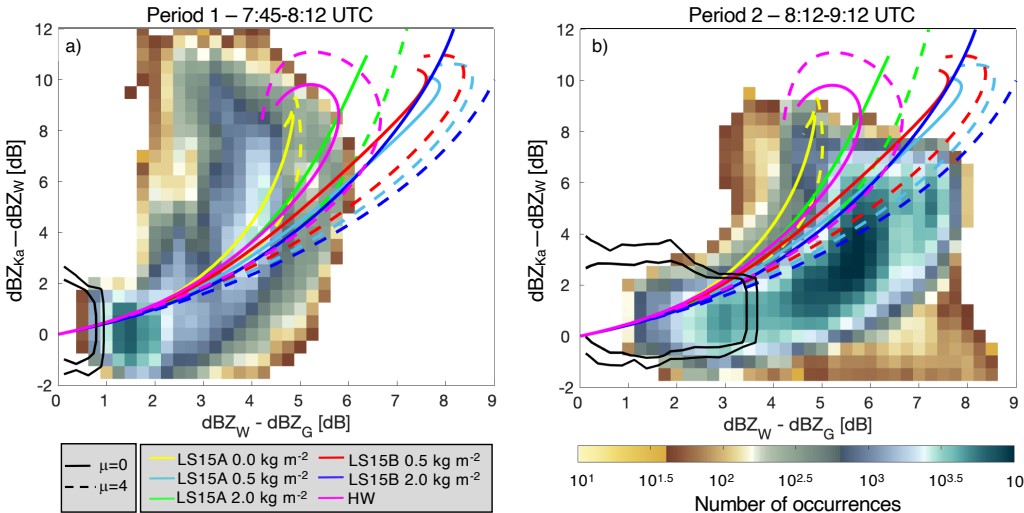

**Figure 7: For observations collected a) between 7:45–8:12 UTC and b) between 8:12–9:12 UTC; distribution of Ka-W dual-wavelength ratio as a function of W-G dual-wavelength ratio for the cloud region between 2 and 5.5 km altitude (colormap) and for the cloud region between 5.75 and 7 km altitude (contours). Lines represent effective reflectivity calculated using scattering models with different particle type (colors) and with different particle size distribution shape parameter (line type). More details about these scattering models is given in the text.**



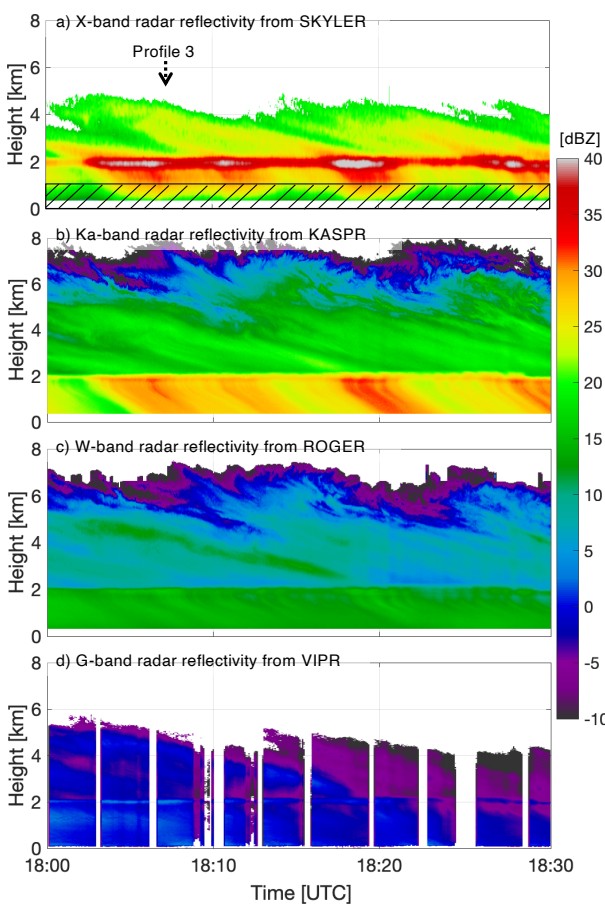

**Figure 8: Time-height of radar reflectivity measured by the a) SKYLER, b) KASPR, c) ROGER and d) VIPR between 18:00 and 18:30 UTC on 25 February 2020. Observations covered by the hashed region in (a) are known to be biased low because of a human error in setting the radar receiver blanking parameters. The arrow in (a) points to the time of the profile displayed in Fig. 9.**





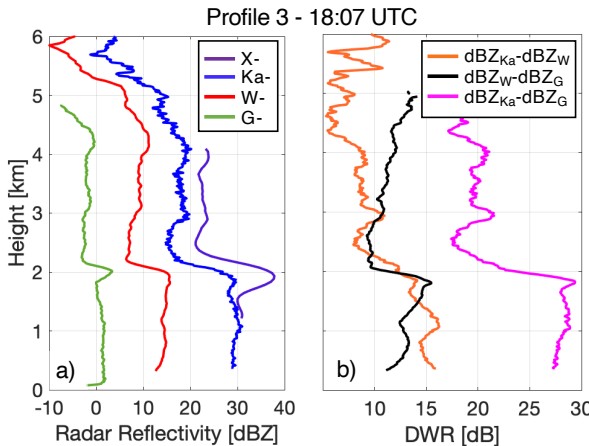

**Figure 9: Profiles taken at 18:07 UTC during the rain period, a) radar reflectivity measured by VIPR (G-band, green), ROGER (W-band, red), KASPR (Ka-band, blue) and SKYLER (X-band, purple) and b) associated dual-wavelength ratio from the Ka-W pair (orange), the W-G pair (black) and the Ka-G pair (magenta).**

