# Peer review of "Multi-frequency Radar Observations of Clouds and Precipitation Including the G-band"

_Atmospheric Measurement Techniques, 2020_

## Referee Comment (RC1) · Anonymous Referee #1 · 20 Jan 2021

The paper presents the first observations of clouds and precipitation with a G-band (167 GHz) radar in combination with radars at lower frequencies. The authors present initial results from the radar collected in both icy precipitation and in rain. They conclude with recommendations for further applications of G-band radars in atmospheric science.

The paper is a useful contribution as a first demonstration of a new techonology for atmospheric observations. As the authors themselves note, this does not purport to be a comprehensive study as it presents a rather limited dataset. The presentation of the results is straightforward and clear. I only have some fairly minor comments that should be considered before final publication.

Title: Maybe this should mention that this is a cloud/precipitation application? For

example, "First Light Multi-Frequency Observations of clouds and precipitation with a G-band radar". Yes, it makes it a bit more wordy but G-band radars have been used in other contexts before.

Line 17: "scatters" -> "scatterers"

Lines 97-98: It would be nice to mention the location of the SBRO here also in more intuitive terms (i.e. New York state, USA).

Line 124: "due to increased atmospheric absorption": this sentence is not quite clear, is there increased absorption on the 167 GHz channel or on the other channel that was not used (I assume the latter)?

Line 130: You mention here when the VIPR was removed but I don't see it mentioned anywhere when the observations started.

Line 145: "spectra" should be singular "spectrum"

Lines 236-238: Besides points 1-3, what about the microphysical changes between the lowest radar-observed altitude and the surface?

Line 252: Is there an explanation for the rather large coefficient for SKYLER?

Line 309: "While Ka- and W-band signals lack sufficient differential scattering to gain further information about such small ice crystals, our observations suggest that G-band signal can": This doesn't make grammatic sense, please rephrase

Line 317: "converge" -> "convergence"

Lines 317-318: "considerable water condensate mass in the atmospheric column": probably, but what about attenuation by the icy hydrometeors?

Lines 343-345: Has the self-similar Rayleigh-Gans approximation been validated for G-band? If so, references should be added. This frequency range has been explored relatively little with models so one should be a bit careful before trusting the results.

[Figure]

Line 355: It seems to me that even a 0.5 dB shift could align the density maximum with the yellow line quite well.

Line 358: "This suggests that the particles observed are not represented the scattering libraries used and calls for further research.": I agree with the "calls for further research" part but I think it would be appropriate to consider other causes before declaring that the scattering libraries are at fault. The deviation of the data from the theoretical curves is not that large, so it could be explained by e.g. remaining calibration bias or incorrect assumptions about the particle size distribution.

Lines 359, 364-365: Perhaps it would be worth pointing out that unrimed particles are consistent with the lack of cloud liquid water that you find.

Lines 363-364: Can the estimate about the fall speed be confirmed by the Doppler? The gravity wave pattern may complicate things but maybe you could e.g. average over one period of the wave.

Line 373: The fall speed here would suggest more rimed particles, again consistent with having more liquid water.

Line 423: "Longer frequencies" -> "longer wavelengths"?

Lines 466-468: The Rayleigh-plateau method is only useful for ground-based studies, right? Not e.g. airborne radars.

Line 477: There is more differential signal, but wouldn't the increased uncertainty in attenuation be a limiting factor?

Line 489: See my comment for line 358.

Figure 8: The color map you are using in this plot is creating false contours. For example in panel c near the melting layer, the reflectivity transitions from light blue to light green through dark green. This creates a narrow band of dark green that falsely looks like a very narrow peak. This effect is one of the reasons why the scientific

community tends to be moving away from the rainbow-scale color maps. You may want to consider doing this plot and panels a-c of Fig. 4 using a perceptually uniform colormap.

---

## Referee Comment (RC2) · Anonymous Referee #2 · 18 Feb 2021

Since Battaglia et al. (2014) there is a lot of excitement about prospects of using G-band radars for cloud studies. This study shows the first comparison of G-band radar observations to X, Ka and W-band radar observations. It is shown that the current technology allows to achieve measurements that are useful for cloud studies. The results are very promising. I find the manuscript to be clearly written. I have a few comments on how the readability of the text can be improved. I also have a couple comments on the reached conclusions and would like to see authors' response to these.

**Comments: Lines 352-353:** "The first period (7:45 − 8:12 UTC) depicted in Fig. 7a corresponds to the period we established had little to no liquid water and presented a high DWR slanted feature (referring back to Fig. 5)." I am not sure how this conclusion

was made. The lidar observations in Fig. 5 indicates presence of two liquid layers at this time, which you point out on line 285. These layers are not very optically thin but may affect the attenuation.

**Line 358:** "This suggests that the particles observed are not represented the scattering libraries used and calls for further research." This conclusion is not necessarily correct. The PSD of snow aggregates tend to be super exponential (Westbrook et al. 2004), i.e. the shape parameter is negative. The super exponential PSD will push the triple frequency curve to the left (Mason et al. 2019), so even for the given scattering models you may be able to reproduce the observations. Westbrook, C. D., Ball, R. C., Field, P. R., and Heymsfield, A. J. (2004), Universality in snowflake aggregation, Geophys. Res. Lett., 31, L15104, doi:10.1029/2004GL020363.

Mason, S. L., Hogan, R. J., Westbrook, C. D., Kneifel, S., Moisseev, D., and von Terzi, L., 2019: The importance of particle size distribution and internal structure for triple-frequency radar retrievals of the morphology of snow, Atmos. Meas. Tech., 12, 4993–5018, https://doi.org/10.5194/amt-12-4993-2019, 2019

**Lines 399-401**: "In the non-Rayleigh scattering regime, $\sigma_b$ does not monotonically increase with $D^6$ but rather follows a lower power of quasi-periodic form with exponential damping of the oscillation (Fig. 4 of Kollias et al., (2007a))."

Are you describing the resonance scattering regime, or as sometime referred to as Mie scattering? If yes, just say that.

**Line 405:** The sentence starting as "Previous work has associated the top boundary..." is, in my opinion too long, and a bit difficult to follow. It would help if you could simplify it.

There are several new studies discussing how ML boundaries depend on radar frequency:

Li, H., and D. Moisseev, 2020: Two layers of melting ice particles within a single radar bright band: Interpretation and implications. Geophys. Res. Lett., 47, e2020GL087499. https://doi.org/10.1029/2020GL087499

And how ML radar signatures at different wavelengths depend on snow properties:

Li, H., Tiira, J., von Lerber, A., and Moisseev, D., 2020: Towards the connection between snow microphysics and melting layer: insights from multifrequency and dual-polarization radar observations during BAECC, Atmos. Chem. Phys., 20, 9547–9562, https://doi.org/10.5194/acp-20-9547-2020.

**Line 433:** "The other fact that SKYLER could also not observe the cloud top also speaks to the importance of operating sensitive X-band radars for cloud studies (liquid attenuation not being an issue at X-band)." You may want to generalize this statement to cm-wavelength (i.e. Ku-band or C-band) radars that are not suffering from significant attenuation as well.

**Conclusions:**

**Line 459, point 2:** While I agree with this conclusion, I miss a discussion in the results section that supports this conclusion. If it is not there, you may want to include it.

**Line 466, point 3:** While high sensitivity is important and you demonstrate that it is possible to achieve it, whether the Rayleigh plateau will be reached will also depends on attenuation. Therefore, it would limit this application to relatively optically thin clouds. The -20 dBZ requirement, as far as I remember, originates from one of Hogan's studies and is referring to unattenuated reflectivity. You should point it out in the discussion.

**Line 487, point 7:** I think this conclusion is not well supported. In addition to what I said above, you only have tested one single scattering library.

---

## Author Comment (AC1) · 24 Mar 2021

The authors would like to thank the reviewers for their excellent suggestions. A point-by-point response to their constructive comments is provided below.

**Reviewer 1:**

Title: Maybe this should mention that this is a cloud/precipitation application? For C1 example, "First Light Multi-Frequency Observations of clouds and precipitation with a G-band radar". Yes, it makes it a bit more wordy but G-band radars have been used in other contexts before.

We agree with the reviewer that the manuscript could benefit for a more precise title. We propose changing the title to "Multi-frequency Radar Observations of Clouds and Precipitation Including the G-band"

**Line 17: "scatters" -> "scatterers"**

This spelling error was corrected.

**Lines 97-98: It would be nice to mention the location of the SBRO here also in more intuitive terms (i.e. New York state, USA).**

Excellent suggestion. The text was revised accordingly. "The SBRO is a fenced-in facility on the edge of Stony Brook University's commuter parking lot located on Long Island, New York, USA (40°53'50" N, 73°07'38" W)."

**Line 124: "due to increased atmospheric absorption": this sentence is not quite clear, is there increased absorption on the 167 GHz channel or on the other channel that was not used (I assume the latter)?**

We agree that this sentence was poorly constructed. It was rephrased to improve clarity. "For the multi-frequency analysis in this work, we only focus on the measurements at 167 GHz since it experiences less gas absorption than VIPR's higher frequency channel."

**Line 130: You mention here when the VIPR was removed but I don't see it mentioned anywhere when the observations started.**

Upon reviewing the manuscript, we realized that the information about VIPR deployment was a bit scattered. We revised the manuscript to address this issue.

The beginning of Section 2 still states "This section provides details specific to the operation of these systems during the deployment of the G-band VIPR radar on February 25th, 2020 [...]"

Section 2.1 now reads: "Around noon on Feb. 24, 2020 (one day before the official field deployment) VIPR was installed near, but outside a large shipping container. That day VIPR was mostly operated off zenith for calibration purposes (details in Sect. 3.2). On the

official deployment day of Feb. 25, 2020, VIPR continued operating next to the large shipping container but this time in vertically pointing mode. Following the onset of rain that day, VIPR's transmitter had to be turned off on a number of occasions to wipe water droplets off of the radar antenna (gaps seen in Fig. 8c). In some instances, we noted that strong radar returns from close-range rain caused an increase in the system noise floor of up to 20 dB stemming from broadband phase noise in the transmitted signal (Cooper et al., 2020). At 20:41 UTC, following the onset of heavier surface rain, VIPR was moved inside the adjacent container and pointed 40° off zenith. Note that off-zenith observations collected during the official deployment were not analyzed as part of the current study."

**Line 145: "spectra" should be singular "spectrum"**

The word "spectra" was replaced by "spectrum" everywhere it is used.

**Lines 236-238: Besides points 1-3, what about the microphysical changes between the lowest radar-observed altitude and the surface?**

The reviewer makes a good point. We have modified the text to reflect this point.

"The several hundred-meter path between these measurements results in three sources of systematic calibration error that can be addressed: 1) radar signal attenuation by atmospheric gases present in the path, 2) radar signal attenuation by the raindrops present in the path and 3) a time lag reflecting the time it takes raindrops to fall from the observed height to the surface. Changes in the particle size distribution due to processes like evaporation and collision/coalescence may also occur but since these changes are nearly impossible to quantify, they remain a source of uncertainty. The first 3 effects can be corrected for [...]"

**Line 252: Is there an explanation for the rather large coefficient for SKYLER?**

This experiment was SKYLER's first "official" deployment. Before then, it had never better carefully calibrated.

**Line 309: "While Ka- and W-band signals lack sufficient differential scattering to gain further information about such small ice crystals, our observations suggest that G-band signal can": This doesn't make grammatic sense, please rephrase**

The end of this paragraph was re-written to improve readability:

"In this region, absence of Ka-W differential signal (i.e.,  $DWR = 0 \, dB$ ) suggests the presence of Rayleigh targets (Tridon et al., 2020). At these frequencies Rayleigh targets correspond to ice populations with PSDs of mass-weighted mean diameter smaller than ~1 mm (Tridon et al., 2019). The absence of differential scattering signals at Ka-W band prevents us from gaining further information about such small ice crystals. On the other hand, the presence of a differential signal at Ka-G band and W-G band of the order of a few decibels across most the layer (see Fig 5b-c and Fig. 6b) suggests that DWR estimates

that use G-band signals can provide size information about smaller ice crystals."

**Line 317: "converge" -> "convergence"**

This spelling error was corrected.

**Lines 317-318: "considerable water condensate mass in the atmospheric column": probably, but what about attenuation by the icy hydrometeors?**

We agree with the reviewer that both liquid and ice could contribute to the observed attenuation; that is already what we intended to convey using the word "water condensate". As to eliminate any ambiguity, we now write:

"The lack of convergence at 0 dB in the profile extracted at 8:42 UTC suggests the presence of considerable water condensate (liquid and/or ice) mass in the atmospheric column (Fig. 6d)."

**Lines 343-345: Has the self-similar Rayleigh-Gans approximation been validated for Gband? If so, references should be added. This frequency range has been explored relatively little with models so one should be a bit careful before trusting the results.**

We agree with the reviewer that the scattering calculations presented in this study are not comprehensive and as such only offer a first idea at the performance of a subset of existing formulations. The revised manuscript was revised to additionally include results from Discrete Dipole Approximation calculations and the text was revised to better outline the scope of the current study.

"Overlaid are DWRKa-W-DWRW-G estimated using self-similar-Rayleigh-Gans approximation and different particle type models and PSD; specifically, unrimed aggregates are represented using the mass-diameter relationships from Hogan and Westbrook (2014) (hereafter HW14) and that of Leinonen and Szyrmer (2015) (hereafter LS15) particle class A. Rimed aggregates are represented using the mass-diameter relationships of LS15 for particle type B with 2 kg m-2 of liquid water path. Also overlaid are DWRKa-W-DWRW-G estimated using Discrete Dipole Approximation scattering calculations for different particle types following formulation prepared by Eriksson et al., (2018) (hereafter E19); specifically: icon graupel, block column, plate, sector snowflake and flat three bullet rosette. Since the shape of the PSD may also impact the scattering of the ice crystal population, PSDs are represented using a gamma function with a shape parameter ( $\mu$ ) of either 0 or 4. We acknowledge that this does not encompass all PSD shapes such as the super exponential one of aggregate populations reported by Westbrook et al. (2004). In any case, the idea is to use overlap between the observed and estimated DWR-DWR to gain information about particle habit."

"Figure 7: For observations collected a) between 7:45–8:12 UTC and b) between 8:12– 9:12 UTC; distribution of Ka-W dual-wavelength ratio as a function of W-G dualwavelength ratio for the cloud region between 2 and 5.5 km altitude (colormap) and for the cloud region between 5.75 and 7 km altitude (contours). Lines represent effective reflectivity calculated using scattering models with different particle type (colors) and with different particle size distribution shape parameter (line type). More details about these scattering models are given in the text."

When it comes to the self-similar Rayleigh-Gans approximation in particular, it scales with the size parameter, so in principle it should be applicable to compute ice scattering even at higher frequencies especially considering that the refractive index of ice is not changing much with frequency. Of course, validating these scattering estimates is a complex task. This task partly motivated the deployment of G-band radars and of multi-frequency radar observatories, which, with their active measurements, are better suited than passive radiometric sensors to constrain ice microphysical retrievals.

**Line 355: It seems to me that even a 0.5 dB shift could align the density maximum with the yellow line quite well.**

Line 358: "This suggests that the particles observed are not represented the scattering libraries used and calls for further research.": I agree with the "calls for further research" part but I think it would be appropriate to consider other causes before declaring that the scattering libraries are at fault. The deviation of the data from the theoretical curves is not that large, so it could be explained by e.g., remaining calibration bias or incorrect assumptions about the particle size distribution.

Following the reviewer's comments, this portion of the manuscript was rephrased.

"For this particular period, a 0.5 dB offset is seen suggesting that a slight adjustment should be made to the observed DWR before they can be interpreted in terms of differential scattering and used to infer particle habit. Even with this slight adjustment, we find that the scattering calculation results only partially match the DWR-DWR signatures observed leaving a noticeable gap in the high (> 7 dB) DWRKa-W and low (< 5 dB) DWRW-G region. This gap could result from outstanding radar calibration bias or from a misrepresentation of the particle size distribution and/or shape of naturally occurring ice crystal in existing scattering libraries. In any case, it calls for further research. We note that the scattering models that are closest to the observed values are those for unrimed aggregates (yellow and magenta lines) and plates (cyan line). "

**Lines 359, 364-365: Perhaps it would be worth pointing out that unrimed particles are consistent with the lack of cloud liquid water that you find.**

Great suggestion by the reviewer. The text was revised accordingly.

"These low fall speeds would be consistent with the presence of unrimed particles; something that is also in line with our conclusion that this period did not present significant amounts of supercooled liquid. Altogether the large DWR values and the slow terminal velocity suggest the presence of large and fluffy, unrimed particles (Locatelli and Hobbs, 1974)."

**Lines 363-364: Can the estimate about the fall speed be confirmed by the Doppler? The gravity wave pattern may complicate things but maybe you could e.g. average over one period of the wave.**

Great suggestion by the reviewer. We analyzed the KASPR Doppler spectra to gain further insight into the fall speed of the ice crystal populations observed during the two periods of interest.

For the first period: "Based on the velocity of the primary peak in the KASPR Doppler spectra over the period, we estimate the fall speed of the ice particles to be roughly 0.8 m s-1."

For the second period: "Based on the velocity of the primary peak in the KASPR Doppler spectra over the period, we estimate the fall speed of the ice particles to be roughly 1.3 m  $s^{-1}$ ."

**Line 373: The fall speed here would suggest more rimed particles, again consistent with having more liquid water.**

Good point. The text was revised accordingly.

"Such faster fall speeds would be consistent with the presence of rimed particles; something that is also in line with our conclusion is that this period presented significant amounts supercooled liquid."

**Line 423: "Longer frequencies" -> "longer wavelengths"?**

Of course. This oversight was corrected in the revised manuscript.

**Lines 466-468: The Rayleigh-plateau method is only useful for ground-based studies, right? Not e.g. airborne radars.**

Excellent question by the reviewer. In theory the Rayleigh-plateau method can be applied to airborne radar data. If vertically pointing, the principles described in this study apply as is. If downward pointing, one would need to be able to detect Rayleigh scatterers near cloud base instead of cloud top, given they are present. Alternatively, downward pointing radars can exploit Earth surface signals for referencing just like a number of spaceborne radars algorithms do.

**Line 477: There is more differential signal, but wouldn't the increased uncertainty in attenuation be a limiting factor?**

The differential signal is the sum of differential scattering and absorption. Where the absorption is from gases and liquid, the use of soundings (for gases) and of the Rayleigh plateau (for liquid) can be applied to all millimeter-wavelength combination to separate the scattering and absorption signal with similar levels of confidence. Once scattering signals have been isolated, the fact that the scattering signal tends to be larger for Ka-G frequency pair should provide increased particle size retrieval confidence.

**Line 489: See my comment for line 358.**

Following the reviewer's comments, this portion of the manuscript was rephrased.

"The scattering libraries tested could only provide a partial explanation of the scattering properties of the ice crystals observed with gaps in the high (> 7dB) DWRKa-W and low (< 5 dB) DWRW-G region. This gap could result from outstanding radar calibration bias, or from a misrepresentation of the particle size distribution and/or shape of naturally occurring ice crystal; in any case additional triple frequency observations including G-band would help confirm this finding, which, if correct, should motivate further research into the scattering properties of naturally occurring ice crystal populations."

Figure 8: The color map you are using in this plot is creating false contours. For example in panel c near the melting layer, the reflectivity transitions from light blue to light green through dark green. This creates a narrow band of dark green that falsely looks like a very narrow peak. This effect is one of the reasons why the scientific community tends to be moving away from the rainbow-scale color maps. You may want to consider doing this plot and panels a-c of Fig. 4 using a perceptually uniform colormap.

The authors maintain that the colorbar used does not misrepresent their findings. Actually,

we would like to argue that the use of colors helps narrow down the range of reflectivity observed in different parts of the system, something that would be difficult to grasp in a black and white image. For the reviewer's benefit we have recreated Fig. 8 in black and white. Contrasting the original color figure and the black and white figure it is evident that the narrow band feature near 2 km is real and not a result of the colorbar choice.

---

## Author Comment (AC2) · 24 Mar 2021

The authors would like to thank the reviewers for their excellent suggestions. A point-by-point response to the comments is provided below.

**Reviewer 2:**

*Comments:*

**Lines 352-353: "The first period (7:45 – 8:12 UTC) depicted in Fig. 7a corresponds to the period we established had little to no liquid water and presented a high DWR slanted feature (referring back to Fig. 5)." I am not sure how this conclusion was made. The lidar observations in Fig. 5 indicates presence of two liquid layers at this time, which you point out on line 285. These layers are not very optically thin but may affect the attenuation.**

> We have revised the text to address the reviewers concerns regarding a lack of clarity.

> "The first period (7:45 – 8:12 UTC) depicted in Fig. 7a corresponds to the period that presented a high DWR slanted feature (referring back to Fig. 5) and a thin liquid layer (referring back to the lidar backscatter observations of Fig. 4). Plotting the radar observations in DWR-DWR space can help determine if the amount of liquid attenuation caused by this thin liquid layer is significant thus preventing us for inferring particle habit directly from the gas attenuation corrected and calibrated radar measurements. To be exact, a clustering of the DWR-DWR observations collected in the upper part of the cloud (between 5.75-7.00 km) near the 0,0 point (depicted by the contours on Fig. 7a) would indicate an absence of signal attenuation. For this particular period, a 0.5 dB offset is seen suggesting that a slight adjustment should be made to the observed DWR before they can be interpreted in terms of differential scattering and used to infer particle habit."

**Line 358: "This suggests that the particles observed are not represented the scattering libraries used and calls for further research." This conclusion is not necessarily correct. The PSD of snow aggregates tend to be super exponential (Westbrook et al. 2004), i.e. the shape parameter is negative. The super exponential PSD will push the triple frequency curve to the left (Mason et al. 2019), so even for the given scattering models you may be able to reproduce the observations.**

**Westbrook, C. D., Ball, R. C., Field, P. R., and Heymsfield, A. J. (2004), Universality in snowflake aggregation, Geophys. Res. Lett., 31, L15104, doi:10.1029/2004GL020363.**

**Mason, S. L., Hogan, R. J., Westbrook, C. D., Kneifel, S., Moisseev, D., and von Terzi, L., 2019: The importance of particle size distribution and internal structure for triple-frequency radar retrievals of the morphology of snow, Atmos. Meas. Tech., 12, 4993– 5018, https://doi.org/10.5194/amt-12-4993-2019, 2019**

> We agree with the reviewer that our original statement was too definitive and modified the text to acknowledge that other factors would need to be explored to get a more comprehensive picture of the accuracy of all existing scattering libraries. We also now include scattering calculations made using the Discrete Diploe Approximation to provide a perspective other than the self-similar-Rayleigh-Gans approximation.

"Overlaid are $DWR_{Ka-W}$-$DWR_{W-G}$ estimated using self-similar-Rayleigh-Gans approximation and different particle type models and PSD; specifically, unrimed aggregates are represented using the mass-diameter relationships from Hogan and Westbrook (2014) (hereafter HW14) and that of Leinonen and Szyrmer (2015) (hereafter LS15) particle class A. Rimed aggregates are represented using the mass-diameter relationships of LS15 for particle type B with 2 kg m$^{-2}$ of liquid water path. Also overlaid are $DWR_{Ka-W}$-$DWR_{W-G}$ estimated using Discrete Dipole Approximation scattering calculations for different particle types following formulation prepared by Eriksson et al., (2018) (hereafter E19); specifically: icon graupel, block column, plate, sector snowflake and flat three bullet rosette. Since the shape of the PSD may also impact the scattering of the ice crystal population, PSDs are represented using a gamma function with a shape parameter (μ) of either 0 or 4. We acknowledge that this does not encompass all PSD shapes such as the super exponential one of aggregate populations reported by Westbrook et al. (2004). In any case, the idea is to use overlap between the observed and estimated DWR-DWR to gain information about particle habit.

The first period (7:45 – 8:12 UTC) depicted in Fig. 7a corresponds to the period that presented a high DWR slanted feature (referring back to Fig. 5) and a thin liquid layer (referring back to the lidar backscatter observations of Fig. 4). Plotting the radar observations in DWR-DWR space can help determine if the amount of liquid attenuation caused by this thin liquid layer is significant thus preventing us for inferring particle habit directly from the gas attenuation corrected and calibrated radar measurements. To be exact, a clustering of the DWR-DWR observations collected in the upper part of the cloud (between 5.75-7.00 km) near the 0,0 point (depicted by the contours on Fig. 7a) would indicate an absence of signal attenuation. For this particular period, a 0.5 dB offset is seen suggesting that a slight adjustment should be made to the observed DWR before they can be interpreted in terms of differential scattering and used to infer particle habit. Even with this slight adjustment, we find that the scattering calculation results only partially match the DWR-DWR signatures observed leaving a noticeable gap in the high (> 7 dB) $DWR_{Ka-W}$ and low (< 5 dB) $DWR_{W-G}$ region. This gap could result from outstanding radar calibration bias or from a misrepresentation of the particle size distribution and/or shape of naturally occurring ice crystal in existing scattering libraries. In any case, it calls for further research. We note that the scattering models that are closest to the observed values are those for unrimed aggregates (yellow and magenta lines) and plates (cyan line)."

[Figure]

"Figure 7: For observations collected a) between 7:45–8:12 UTC and b) between 8:12–9:12 UTC; distribution of Ka-W dual-wavelength ratio as a function of W-G dual-wavelength ratio for the cloud region between 2 and 5.5 km altitude (colormap) and for the cloud region between 5.75 and 7 km altitude (contours). Lines represent effective reflectivity calculated using scattering models with different particle type (colors) and with different particle size distribution shape parameter (line type). More details about these scattering models are given in the text."

**Lines 399-401: "In the non-Rayleigh scattering regime, $\sigma_b$ does not monotonically in- crease with $D^6$ but rather follows a lower power of quasi-periodic form with exponential damping of the oscillation (Fig. 4 of Kollias et al., (2007a))." Are you describing the resonance scattering regime, or as sometime referred to as Mie scattering? If yes, just say that.**

The reviewer is correct. The sentence was rephased to improve the reference to this known scattering behavior.

"In the non-Rayleigh scattering regime, $\sigma_b$ does not monotonically increase with $D^6$ but rather follows a lower power resonance pattern with damping of the oscillation (Fig. 4 of Kollias et al., (2007a))."

**Line 405: The sentence starting as "Previous work has associated the top boundary..." is, in my opinion too long, and a bit difficult to follow. It would help if you could simplify it. There are several new studies discussing how ML boundaries depend on radar frequency:**

**Li, H., and D. Moisseev, 2020: Two layers of melting ice particles within a single radar bright band: Interpretation and implications. Geophys. Res. Lett., 47, e2020GL087499. https://doi.org/10.1029/2020GL087499**

**And how ML radar signatures at different wavelengths depend on snow properties:**

**Li, H., Tiira, J., von Lerber, A., and Moisseev, D., 2020: Towards the connection be- tween snow microphysics and melting layer: insights from multifrequency and dual- polarization radar observations during BAECC, Atmos. Chem. Phys., 20, 9547–9562, https://doi.org/10.5194/acp-20-9547-2020.**

We would like to thank the reviewer for bringing to our attention these two very recent publications. Upon reading these 2 articles we have revised our discussion of the bright band signature.

"Inferring information about the ice melting process from the properties of the radar-detected bright band is still an active area of research (e.g., Heymsfield et al., 2015;Li et al., 2020). The early work of Fabry and Zawadzki (1995) suggested that the magnitude and vertical extent of the radar reflectivity enhancement at cm-wavelength are influenced by precipitation rate, phase transitions (i.e., liquid coating ice), change in fall speed throughout melting, precipitation growth and changes in the particle size distribution linked to aggregation and breakup. More recent studies using cm-wavelength radars suggested that the depth of the radar bright band, at cm-wavelengths, may be linked to the presence of rimed particles (e.g., Kumjian et al., 2016;Wolfensberger et al., 2016). In contrast, at mm-wavelength radars, non-Rayleigh scattering reduces the influence of large melting snowflakes in determining the magnitude and vertical extent of the melting layer radar signature (Kollias and Albrecht, 2005). In addition, due to their increased relative sensitivity to small melting ice crystals, millimiter-wavelength radars like KASPR and ROGER observe a higher top boundary of their bright band. While not observed here, it has been suggested that W-band radars can provide insight into the activity of the aggregation process because this process is believed to cause of a dip, as opposed to the enhancement that is the bright band, in the radar reflectivity profile (a.k.a. dark band; (Sassen et al., 2005;Sassen et al., 2007;Heymsfield et al., 2008)). Interestingly, observations collected by the VIPR reveal a well-defined bright band at G-band frequency. VIPR's bright band differs from that of the other radars in two main ways: 1- its top boundary is slightly higher compared to that of the W-band, 2- its bottom boundary is higher than that of the X-band. These discrepancies are in line with our interpretation that VIPR's signal is controlled by the melting of even smaller ice crystals. This agrees with Li and Moisseev (2020) interpretation  that the radar bright band properties depend on the radar wavelength since the radar wavelength effectively dictates the ice population size "in focus"."

**Line 433: "The other fact that SKYLER could also not observe the cloud top also speaks to the importance of operating sensitive X-band radars for cloud studies (liquid attenuation not being an issue at X-band)." You may want to generalize this statement to cm-wavelength (i.e. Ku-band or C-band) radars that are not suffering from significant attenuation as well.**

Good suggestion by the reviewer. The sentence was revised accordingly.

"The other fact that SKYLER could also not observe the cloud top also speaks to the importance of operating sensitive X-band radars for cloud studies (liquid attenuation not

being an issue at cm wavelengths)."

*Conclusions:*

**Line 459, point 2: While I agree with this conclusion, I miss a discussion in the results section that supports this conclusion. If it is not there, you may want to include it.**

>This conclusion emerged from our gas attenuation correction activity described in Sec. 3.1 and presented below for reference.
>
>"For this particular mid-latitudinal winter case, we estimate two-way gas attenuation at 11 km to reach ~0.1 dB at X-band, ~0.5 dB at Ka-band, ~2.0 dB at W-band and 10.0 dB at G-band. The large variability in gas attenuation from frequency to frequency, especially near water vapor absorption lines, is what allows DAR techniques to be used for water vapor profiling. On the upside, the notable magnitude of the gas attenuation at higher-frequencies (i.e., W-band but even more so G-band) makes them ideal frequencies to use for such application. On the downside, significant gas attenuation hinders the sensitivity of high frequency radars to clouds and light precipitation."

**Line 466, point 3: While high sensitivity is important and you demonstrate that it is possible to achieve it, whether the Rayleigh plateau will be reached will also depends on attenuation. Therefore, it would limit this application to relatively optically thin clouds. The -20 dBZ requirement, as far as I remember, originates from one of Hogan's studies and is referring to unattenuated reflectivity. You should point it out in the discussion.**

>The reviewer makes a very good point. The discussion related to point 3 was expanded to touch on this important point.
>
>"Nominally radar systems should be capable of detecting unattenuated reflectivity as weak as -40 dBZ at 1 km after 1-s signal integration (i.e., -20 dBZ at 10 km altitude). In the present study, the radars deployed generally meet this sensitivity criteria. It follows that deployments in humid environments would drive higher sensitivity requirements because of enhanced signal attenuation by water vapor. The same can be said about deployments in liquid containing clouds where enhanced signal attenuation by liquid water is to be expected."

**Line 487, point 7: I think this conclusion is not well supported. In addition to what I said above, you only have tested one single scattering library.**

>Again, we agree with the reviewer that our original statement was too definitive and modified the text to acknowledge that other factors would need to be explored to get a more comprehensive picture of the accuracy of all existing scattering libraries. We also now include scattering calculations made using the Discrete Diploe Approximation to provide a perspective other than the self-similar-Rayleigh-Gans approximation.

"The scattering libraries tested could only provide a partial explanation of the scattering properties of the ice crystals observed with gaps in the high (> 7dB) DWR$_{Ka-W}$ and low (< 5 dB) DWR$_{W-G}$ region. This gap could result from outstanding radar calibration bias, or from a misrepresentation of the particle size distribution and/or shape of naturally occurring ice crystal; in any case additional triple frequency observations including G-band would help confirm this finding, which, if correct, should motivate further research into the scattering properties of naturally occurring ice crystal populations."

[Figure]

"Figure 7: For observations collected a) between 7:45–8:12 UTC and b) between 8:12–9:12 UTC; distribution of Ka-W dual-wavelength ratio as a function of W-G dual-wavelength ratio for the cloud region between 2 and 5.5 km altitude (colormap) and for the cloud region between 5.75 and 7 km altitude (contours). Lines represent effective reflectivity calculated using scattering models with different particle type (colors) and with different particle size distribution shape parameter (line type). More details about these scattering models are given in the text."